# Conceptualization of CO$_2$ Terminal for Offshore CCS Using System Engineering Process

**Hyonjeong Noh [1], Kwangu Kang [1],\*, Cheol Huh [2], Seong-Gil Kang [3], Seong Jong Han [1] and Hyungwoo Kim [1]**

[1] Offshore Industries R&BD Center, Korea Research Institute of Ships & Ocean Engineering, Geoje 53201, Korea; hjnoh@kriso.re.kr (H.N.); sjhan@kriso.re.kr (S.J.H.); hyungwoo4601@kriso.re.kr (H.K.)
[2] Ocean Science & Technology School, Korea Maritime and Ocean University, Busan 49112, Korea; cheolhuh@kmou.ac.kr
[3] Global Cooperation Center, Korea Research Institute of Ships & Ocean Engineering, Daejeon 34103, Korea; kangsg@kriso.re.kr
\* Correspondence: kgkang@kriso.re.kr

**Abstract:** In this study, the basic configuration and operation concept of a CO$_2$ terminal were identified by conducting a system engineering process. The performance goal of a CO$_2$ terminal was determined by requirement analysis. Then, functions and timelines were derived by functional analysis to meet the performance goal. Equipment to perform the functions were defined and finally, a process flow block diagram of the CO$_2$ terminal was acquired. The CO$_2$ terminal in this study consisted of three parts. First, the CO$_2$ loading/unloading part is responsible for liquid CO$_2$ unloading from the carrier and loading vapor CO$_2$ onto the carrier. Secondly, the liquid CO$_2$ transmission part extracts liquid CO$_2$ from the storage tanks and increases the pressure until it satisfies the offshore pipeline transportation condition. The vapor-treatment part collects boil-off gas, generates vapor CO$_2$, and charges the storage tanks with vapor CO$_2$ to control the pressure of the storage tanks that discharge liquid CO$_2$. Finally, the study results were compared with a liquefied natural gas (LNG) terminal. The biggest difference between the CO$_2$ terminal in this study and the LNG terminal is that a vaporizer is essential in the CO$_2$ terminal due to the smaller storage capacity of the CO$_2$ terminal and, therefore, the lower amount of boil-off gas.

**Keywords:** CO$_2$ terminal; CO$_2$ storage tank; system engineering process; conceptual design; CO$_2$ loading/unloading

## 1. Introduction

Many efforts are underway worldwide to reduce atmospheric emissions of carbon dioxide (CO$_2$), which is a dominant cause of global warming. To limit the global temperature increase to below 2 °C, the annual amount of greenhouse gas emissions should be less than approximately 14 Gt CO$_2$ by 2050, which is about a 60% reduction compared with the current level [1]. Countries that have a manufacturing-oriented industrial structure or rely heavily on electricity production from fossil fuels may struggle to quickly switch to a low-carbon emission structure using renewable energy. Carbon capture and storage (CCS) can be an attractive greenhouse gas reduction option for these countries [2]. CCS basically consists of three parts: (1) separating and capturing CO$_2$ from a large-scale emitter such as a power plant; (2) transporting CO$_2$ through pipelines or carriers from capture stations to storage sites; and (3) injecting and storing CO$_2$ into geological formations [3]. Many studies have focused on the capture and storage aspects, but few have focused on transportation [4,5]. However, interest is growing in the transportation aspect as it has been shown to be responsible for a high proportion of the costs in CCS projects [6].

Between pipelines and carriers, which are the two most common methods of $CO_2$ transportation, pipelines have been the most widely used [5]. When transporting $CO_2$ to a pipe, a variable $CO_2$ transportation network is required for the flexible operation of the CCS chain, which is technically challenging [7,8]. Considering the flexibility of $CO_2$ carriers, however, the significance of carriers is likely to increase in the near future [9–13]. A $CO_2$ terminal is required when multiple sources and storage sinks are linked or when pipelines and carriers are linked in the CCS transportation chain [14–17]. In the future, once CCS is widely distributed as a $CO_2$ emission reduction option, the likelihood of CCS projects using multiple sources and storage sites will be high [11]. This means that $CO_2$ terminals will become important facilities within the whole CCS chain that can simultaneously contribute to the realization of $CO_2$ carriers as a rational $CO_2$ transportation option.

Unfortunately, research regarding the configuration, functions, and roles of $CO_2$ terminals has been limited. Only a few studies [15,18–23] addressed $CO_2$ terminals. It is worth mentioning the Norwegian full-scale CCS project named the Northern light project [20,22,24], which includes the concept of $CO_2$ terminal. The terminal is located onshore and has a role of receiving and temporary storing of $CO_2$ transported by a carrier. It is equipped with a vaporizer to maintain vapor and liquid $CO_2$ balance during injection [24]. Vermeulen [15] suggested a $CO_2$ liquid logistics shipping concept for a complex CCS chain connecting multiple $CO_2$ emitters and storage sites. This concept uses complex $CO_2$ terminals that can receive large amounts of $CO_2$ through pipeline networks and barges from various emitters and can send $CO_2$ out to multiple storage sites using carriers and pipelines. However, no study has defined an overall configuration of the terminal and its operational concept. Lee et al. [21] simulated a $CO_2$ process flow in an intermediate storage facility attached to a capture site on the coast for loading $CO_2$ to a carrier. The storage facility in their study consisted of a $CO_2$ input process, a storage tank, a loading process, a recirculation process, and a boil-off gas (BOG) reliquefaction process. However, they did not clearly explain how they chose the components for the $CO_2$ terminal. In a few studies about $CO_2$ carrier transportation, $CO_2$ terminals are considered part of the description of the CCS chain, but the research did not focus on $CO_2$ terminals [18,19,25–27]. Several studies [28–31] focused on the direct injection of $CO_2$ into geological formations from carriers did not consider the $CO_2$ terminal. As such, the configuration of a $CO_2$ terminal, reflecting its functions and roles, has rarely been studied. Conversely, considerable research has been conducted on liquefied natural gas (LNG) terminals [32–43], which can be thought of as similar to $CO_2$ terminals. LNG terminals can be used in developing $CO_2$ terminals, but a cautious approach is required as the transport characteristics of $CO_2$ and LNG differ.

Designing a simple system or a precedent system without systematic design processes is possible. However, for a complex system or an unprecedented system like a $CO_2$ terminal, it is highly recommended to follow a system engineering process (SEP), as shown in Figure 1.

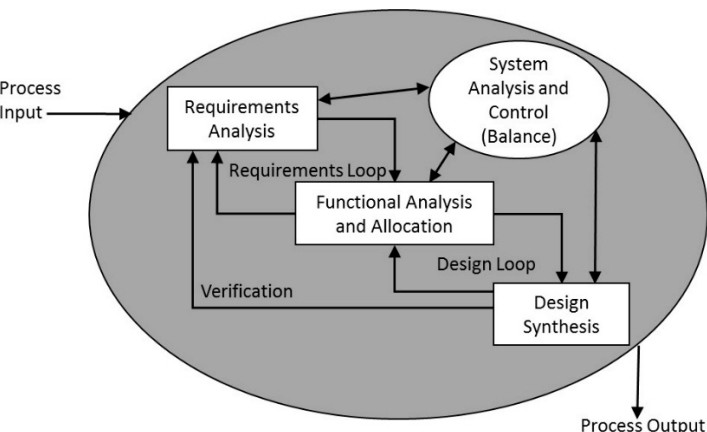

**Figure 1.** The systems engineering process (SEP).

The SEP consists of (1) requirements analysis, which transforms the stakeholder's system needs into functional requirements from an engineering point of view; (2) functional analysis and allocation, which defines what functions are to be performed; and (3) design synthesis, which determines how these functions are combined into the design [44]. The SEP is completed when this series of processes is conducted. The strength of SEP during the concept development stage is that necessary functions are reflected, whereas unnecessary functions are removed in advance.

In this study, the conceptual design of a $CO_2$ terminal was structurally and systematically developed using the SEP. Our conceptual design focuses on the configuration and basic operation concept of a $CO_2$ terminal. In the requirement analysis (Figure 1), a CCS chain was analyzed to derive the goal and performance objectives of a $CO_2$ terminal. For requirement analysis, specific conditions of the CCS demonstration project promoted in Korea were applied. Then, the functions were derived to meet the goal and performance objectives of the $CO_2$ terminal using functional analyses. Several functional analyses were conducted to define the top-level functions and their basic operational concepts. Design synthesis was performed to determine the equipment that corresponds to each function. Based on the SEP, a process flow block diagram of the $CO_2$ terminal was finally suggested that explicitly depicts the configuration of the $CO_2$ terminal.

## 2. Description of CCS Chain

$CO_2$ terminals have various forms depending on the connection methods between capture and storage sites [15]. Since the objective of this study was to demonstrate the basic configuration and operation concept of a $CO_2$ terminal, a relatively simple form of a CCS chain was selected as a target system (Figure 2a). The concept of a CCS chain displayed in Figure 2 is identical to the 1-million-ton-scale demonstration project in Korea. This CCS chain liquefies captured $CO_2$ at a thermal power plant located on the coast, and transports liquid $CO_2$ ($LCO_2$) to a $CO_2$ terminal through $CO_2$ carriers. $CO_2$ is transmitted to an offshore platform through an offshore pipeline and injected into offshore geological formations.

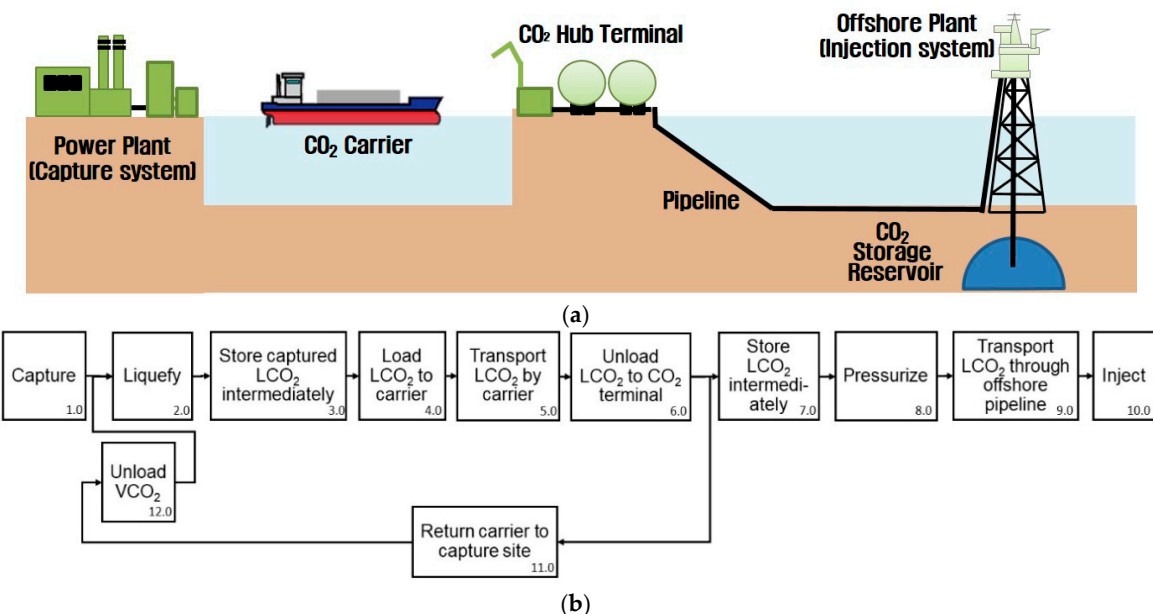

**Figure 2.** The carbon capture and storage (CCS) chain process in this study: (**a**) graphical representation and (**b**) functional flow block diagram. L = liquid, V = vapor.

The CCS chain can be expressed as a functional flow block diagram, as depicted in Figure 2b. Since we focused on the $CO_2$ terminal, we omitted the explanation of a functional flow block diagram of the

whole CCS chain. Contrary to other CCS chain descriptions, the return of $CO_2$ carriers is included to consider the vaporized $CO_2$ ($VCO_2$) in returning carriers.

The specific conditions of the CCS project in Figure 2 are as follows. The annual transport amount is 1 million tons of $CO_2$. The distance from the capture source to the $CO_2$ terminal is 580 km. The temperature and pressure of $CO_2$ are to be maintained at $-27\,°C$ and 16 bar, respectively, during the $CO_2$ carrier and $CO_2$ terminal operation. The $CO_2$ in the terminal is to be pressurized and heated for offshore pipeline transportation.

## 3. System Engineering Process

In this study, the SEP in Figure 1 was followed to identify the basic configuration and operation concept of the $CO_2$ terminal. System engineering is a useful method for designing new systems that are complex or did not exist previously. Figure 1 depicts the system engineering process presented by the U.S. Department of Defense, which is widely used in many fields [44]. The system engineering process involves requirement analysis, functional analysis and design synthesis, as shown in Figure 1. Requirement analysis, functional analysis and allocation, and design synthesis are iterative and mutually complementary [18]. A system analysis and control were performed to balance this series of procedures. A more detailed explanation of each procedure is provided below.

### *3.1. Requirement Analysis*

Requirements analysis is the first step in the system engineering process, in which the system requirements definition process converts the stakeholder representation into a technical representation of the product. The requirement analysis can transforms the project's needs into engineering language, which consequently enables the system designer to conduct the design concept [45]. During requirement analysis, the CCS chain is analyzed to derive the performance requirement of the $CO_2$ terminal.

### *3.2. Functional Analysis*

After identifying the system requirements, functional analysis is performed to define the logical architecture that can satisfy the identified requirements. Functional analysis is the step of defining the basic functions that the system should perform. This analysis focuses on "what" must be performed, not "how" functions will be performed. Notably, this analysis is function-oriented rather than equipment-oriented. In functional analysis, three functional analysis tools are used: a functional flow block diagram (FFBD), integration definition for function modeling (IDEF0), and timeline analysis (TLA). Through FFBD, the functional flow of the whole CCS transport chain is determined. The sequence and relationships between the functions of the $CO_2$ terminal are defined by IDEF0. To consider the time durations of functions, TLA is used. The results of TLA support the operation concept of the $CO_2$ terminal.

### 3.2.1. Functional Flow Block Diagram (FFBD)

FFBD is graphical tool used to show the sequence of all functions that the system should perform. FFBD focuses the sequence of each function, not the time required or the flow of time between functions [8]. In FFBD, each function represented by a block is identified in terms of inputs and outputs [45]. Each block can be expanded to a series of sub-functions. A function is represented by a rectangular block with the title of the function. The title of the function is composed of an action verb followed by a noun. Some functions may be performed in parallel when necessary.

### 3.2.2. Integration Definition for Function Modeling (IDEF0)

The IDEF0 diagram integrates the inputs, control, outputs, and mechanisms of functions. IEDF0 allows us to understand the correlation between functions derived from FFBD. In IDEF0, the block represents the function to be performed, and the left and right arrows show the input and output of

the process, respectively. The up and down arrows indicate the controls and the mechanism of the function, respectively (Figure 3).

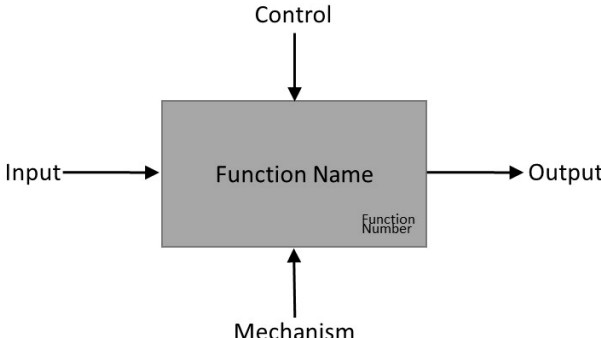

**Figure 3.** Integration definition for function modeling (IDEF0) shows the inputs, control, outputs, and mechanisms of functions.

### 3.2.3. Timeline Analysis (TLA)

Although the FFBD and IDEF0 represent the logical sequence of functions, they cannot show the time duration of or between functions. TLA is a method for identifying specific time-related design or operating requirements, useful for reflecting the durations of time-critical functions, such as reaction time, turnaround time, time limits, etc., and to clarify time-related design constraints. TLA complements FFBD or IDEF0 [46].

### 3.3. Design Synthesis

Physical equipment that performs the functions of the system is specified in the final stage of the design synthesis. In this study, the requirement allocation sheet (RAS) was used for the design synthesis. In the RAS, the configuration or equipment that will perform the functions are derived by connecting the functions with the allocated performance and the physical system. Usually the functions are listed in the left column in the RAS, the performance or requirements that each function should meet are listed in the middle column, and the corresponding equipment that can perform each function is listed in the right column [47]. This methodology clearly indicates the interrelationships between the main equipment and minimizes the risk of missing key design variables.

## 4. Results

### 4.1. $CO_2$ Carrier and $CO_2$ Terminal Requirements

To derive the performance requirements of the $CO_2$ terminal, we clearly defined the logistics concept, including carrier size, number of the carriers, carrier transportation cycle, etc., using the CCS chain conditions of the Korean project described in Section 2. This CCS chain aims to store 1 million tons of $CO_2$ annually, which can be converted into a daily transport rate of 2740 tons/day.

### 4.1.1. Time for Carrier Round Trip and the Availability of Transport

The distance from the capture site to the $CO_2$ terminal is about 580 km, and the carrier speed is usually about 14 knots. In other words, one day is required for one-way transportation. An additional day is required for berthing, purging the pipeline, and loading/unloading $CO_2$ at the capture site and the $CO_2$ terminal each. Therefore, a total of four days is required for one carrier to complete a round trip between the capture site and the $CO_2$ terminal, including loading and unloading of $LCO_2$. The availability of transport is assumed to be less than 80% considering the possibility of bad weather conditions.

### 4.1.2. Number of Carriers

In general, large carriers have lower capital expenditure (CAPEX) and operating expenditure (OPEX) per cargo volume than smaller carriers. Therefore, operating a small number of carriers is cost effective. In other words, the cost of one carrier is inexpensive comparing the cost of two half-sized carriers. A more detailed comparison of costs between one and two carriers is explained in Section 5.3.

When operating only one carrier, if an unexpected failure of the carrier occurs, the entire CCS chain will be stopped, making the continuous injection of $CO_2$ difficult. Intermittent injection of $CO_2$ lowers the $CO_2$ injection capacity, reducing the efficiency of the entire CCS chain [48,49]. On the other hand, if two carriers are operated, one will be able to operate even if the other stops due to an unforeseen failure. Therefore, the entire CCS chain can be operated with a reduced injection rate while avoiding injection interruption, even if it is not able to satisfy the target storage amount of 1 million tons/year. Thus, this study determined the number of carriers as two even though it is inexpensive to operate one carrier. In the case of this study, even if one carrier fails to operate, it will be able to transport 75% of the normal transport rate if the maximum availability of the other carrier is maintained during the repair period.

### 4.1.3. Size of Carrier

With two carriers, four days of shipping time, and less than 80% shipping availability, the minimum amount of $LCO_2$ carried by single transportation can be derived as follows:

$$1 \times 10^6 \text{ tons}/(365 \text{ days} \times 80\%/4 \text{ days})/2 \text{ carriers} \approx 6849 \text{ tons/carrier.} \tag{1}$$

A total of 2.5 days are required to capture 6849 tons of $LCO_2$ considering a capture rate of 2740 tons/day. However, the CCS transport chain can be both safe and simple if one carrier transports 3 days' worth of captured $CO_2$ at one time. Therefore, one carrier transports 8220 tons (2740 tons × 3 days) of $LCO_2$. In this case, the availability is about 66%. Since one carrier transports 3 days' worth of captured $CO_2$, the carrier arrival cycle in the $CO_2$ terminal is also 3 days.

The same volume of $VCO_2$ must be loaded into the carrier's cargo tank when unloading $LCO_2$ from the carrier to the $CO_2$ terminal. The carrier's cargo tank is displaced by the $VCO_2$ of the storage tanks at the terminal, while $LCO_2$ fills the storage tanks. There are two reasons for loading $VCO_2$ into the carrier's cargo tank. The first is to allow the pressure and temperature of the cargo tank to be controlled during the unloading process. Constant pressure and temperature facilitate the process. The second reason is to prevent the rapid decrease in temperature due to Joule–Thomson cooling. This rapid decrease in temperature could lead to material damage [50]. The $VCO_2$ and $LCO_2$ density ratio is about 0.041 under conditions of −27 °C and 16 bar. Therefore, 337 tons of $VCO_2$, which correspond to 4.1% of the 8220 tons of $LCO_2$, is returned to the capture site. For the net transport amount of $CO_2$ to be 8220 tons per carrier, this amount of $CO_2$ should be additionally carried from the capture site. The total transport rate is, therefore, 8557 tons per carrier. Considering the BOG in the cargo tanks during the ship transportation, the cargo tanks should be approximately 95% filled and then, the total size of the $CO_2$ cargo tanks would be increased to be ~9000 tons. In conclusion, two 9000-ton carriers that transport 8857 tons of $LCO_2$ over four days in one shipment are needed in our CCS chain and two unloadings of $CO_2$ in six days would be completed in the $CO_2$ terminal.

### 4.1.4. Size and Number of Storage Tank in Terminal

Since the maximum capacity of a pressurized $CO_2$ storage tank is 5000 tons, the size of the $CO_2$ storage tanks in the $CO_2$ terminal can be easily determined to be half the size of a carrier, 4500 tons. Two empty tanks should be ready before unloading the $LCO_2$. The $CO_2$ terminal should maintain an appropriate amount of buffer $CO_2$ to operate smoothly, although the $CO_2$ supplied by the carrier may be halted for at least 60 h due to typhoons or other events and conditions. The 60-h buffer corresponds to 6850 tons of $CO_2$. At this stage, four $CO_2$ storage tanks are required. The number of storage tanks

will be checked again using TLA to determine if four tanks are sufficient for buffering $CO_2$ in the next section. These requirements will be checked through functional analysis and allocation and design synthesis in the SEP in the following sections.

### 4.1.5. The Estimation of the Amount of BOG in the $CO_2$ Storage Tank at Terminal

To maintain the pressure in the storage tank that emits $LCO_2$ to the pipeline at the $CO_2$ terminal, it is necessary to inject the same volume of $VCO_2$ as the $LCO_2$ is emitted. Since $CO_2$ is stored at a low temperature in the storage tanks, the generation of BOG is inevitable. And in the $CO_2$ terminal design, it is very important to compare the required $VCO_2$ and BOG quantities. In other words, if the amount of BOG generated is less than the required $VCO_2$, a vaporizer is needed, and if it is large, a reliquefier must be installed. Since this study focuses on the concept design of the $CO_2$ terminal, BOG is simply estimated by assuming that the storage tanks are a thin-walled spherical tank of metal covered with a thick insulation layer on the outside as shown in Figure 4. The BOG generated in the storage tanks at the $CO_2$ terminal was determined by assuming that all external heat ingress is converted into latent heat of vaporization of $CO_2$, as shown in the equation below:

$$\dot{m} = \frac{q}{h_{fg}}, \tag{2}$$

where $q$ is heat ingress from the surrounding atmosphere, $\dot{m}$ is vaporized mass of $CO_2$, and $h_{fg}$ is latent heat of vaporization of $CO_2$. Heat ingress is defined as the difference between the ambient air temperature and the temperature of $CO_2$ in the tank divided by the sum of the thermal resistances [51], as shown in Equation (3):

$$q = \frac{T_{\infty,2} - T_{\infty,1}}{R_{tot}}, \tag{3}$$

where $T_{\infty,1}$ and $T_{\infty,2}$ represent the temperature of $CO_2$ in the storage tank and ambient air temperature, respectively. The ambient air temperature, $T_{\infty,2}$ is assumed to be 35 °C, which is the highest temperature in summer in Korea. $R_{tot}$ is the total thermal resistance and consists of the conduction resistances and the convection resistance. The detailed equations are as follows:

$$R_{tot} = R_{cond,\,1} + R_{cond,\,2} + R_{conv} \tag{4}$$

$$R_{cond,\,1} = \frac{1}{4\pi k_1}\left(\frac{1}{r_1} - \frac{1}{r_2}\right) \tag{5}$$

$$R_{cond,\,2} = \frac{1}{4\pi k_2}\left(\frac{1}{r_2} - \frac{1}{r_3}\right) \tag{6}$$

$$R_{conv} = \frac{1}{h4\pi r_3^2} \tag{7}$$

where $R_{cond,1}$ and $R_{cond,2}$ are the conduction resistances in the thin-walled spherical metallic tank whose material is SA537-cl2 and in the insulation layer, respectively. $k_1$ and $k_2$ are the thermal conductivities of SA537-cl2 and insulation material of perlite, whose values are 52 and 0.047 W/mK, respectively. $r_1$, $r_2$, and $r_3$ are inner radius of thin-walled spherical tank, outer radius of thin-walled spherical tank, and radius of insulation outer surface, respectively. The inner radius of the spherical tank, $r_1$, was calculated as 10.24 m to allow the spherical tank to have a volume of 4500 m$^3$. The thickness was calculated to have a range of 39.9 to 44.4 mm by using the American Society of Mechanical Engineers (ASME) Boiler and Pressure Vessel Code Section VIII Division 2 [52]. This study assumes that the thickness of the thin-walled spherical tank is constant to 40 mm and the insulation material layer has a thickness of 0.2 m. $R_{conv}$ is the convection resistance in the outer surface of tank and $h$ is a heat transfer coefficient which is assumed as 20 W/m$^2$K. From Equations (5)–(7), $R_{cond,1}$, $R_{cond,2}$, and $R_{conv}$ were calculated as $5.05 \times 10^{-7}$, $2.73 \times 10^{-3}$, and $3.16 \times 10^{-5}$ K/W, respectively. Since both $R_{cond,1}$ and $R_{conv}$

are much smaller than $R_{cond,2}$, $R_{tot}$ is almost the same as $R_{cond,2}$, where $R_{tot}$ calculated by Equation (4) is $2.77 \times 10^{-3}$ K/W.

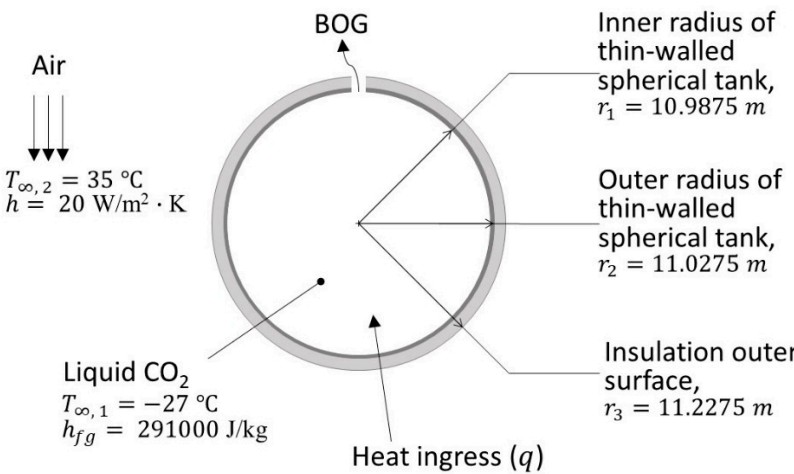

**Figure 4.** Schematics of simplified storage tank for boil-off gas (BOG) calculation.

The amount of BOG calculated by Equation (2) is 241 kg/h. Since the number of tanks in the $CO_2$ terminal is 4, the total BOG is 965 kg/h. The amount of $VCO_2$ required is 4.1% of the emitted $CO_2$, which corresponds to 4680 kg/h. Since the amount of BOG in the storage tanks is about 20% of the required $VCO_2$, a vaporizer is needed to fully obtain the required $VCO_2$ [24]. It is noted that BOG generated per day a tank is 6.65 tons/day, which corresponds to about 0.16% of the $CO_2$ storage capacity. It is also noted that the stratification phenomena inside of the $CO_2$ tank [53] could increase the BOG generation, but the amount of BOG generated would be still smaller than the required amount.

*4.2. Functional Analysis of the $CO_2$ Terminal*

4.2.1. FFBD of $CO_2$ Transport Chain

To derive the basic functions of the $CO_2$ terminal, a functional flow block diagram was used. Since the basic functions should be connected with the entire CCS chain, the three $CO_2$-terminal-related functions—'6.0 Unload $LCO_2$ to $CO_2$ terminal', '7.0 Store $LCO_2$ intermediately', and '8.0 Pressurize' (Figure 2b)—were set as the higher-level functions.

For the higher-level function of '6.0 Unload $LCO_2$ to $CO_2$ terminal', the lower-level functions of '6.1 Unload $LCO_2$' and '6.2 Load $VCO_2$' are developed. The function '6.2 Load $VCO_2$' involves loading the $VCO_2$ into the $CO_2$ cargo tank on the carrier to prevent the rapid pressure decrease of cargo tanks on the carrier caused by unloading $LCO_2$ from the carrier.

To achieve the function of '7.0 Store $LCO_2$ intermediately', $LCO_2$ should be stored in the $CO_2$ storage tank after unloading $LCO_2$ from the carrier at adequate temperature and pressure, and then be transmitted for offshore pipeline transport. Therefore, '7.1 Receive $LCO_2$', '7.2 Store $LCO_2$', and '7.3 Transmit $LCO_2$' functions can be derived as lower-level functions. $VCO_2$, which will be loaded onto the carrier, is supplied from the $CO_2$ terminal, so the function of '7.4 Generate $VCO_2$' can also be derived. The temperature and pressure of $LCO_2$ extracted from the storage tanks needs to be increased to meet the appropriate conditions for injection before transmission through the offshore pipeline. Therefore, the function of '8.1 Increase $LCO_2$ T & P' is included as the lower-level function of '8.0 Pressurize'. To summarize, seven basic functions are defined for the $CO_2$ terminal: '6.1 Unload $LCO_2$', '6.2 Load $VCO_2$', '7.1 Receive $LCO_2$', '7.2 Store $LCO_2$', '7.3 Transmit $LCO_2$', '7.4 Generate $VCO_2$', and '8.1 Increase $LCO_2$ T & P', indicated by the blue region in Figure 5.

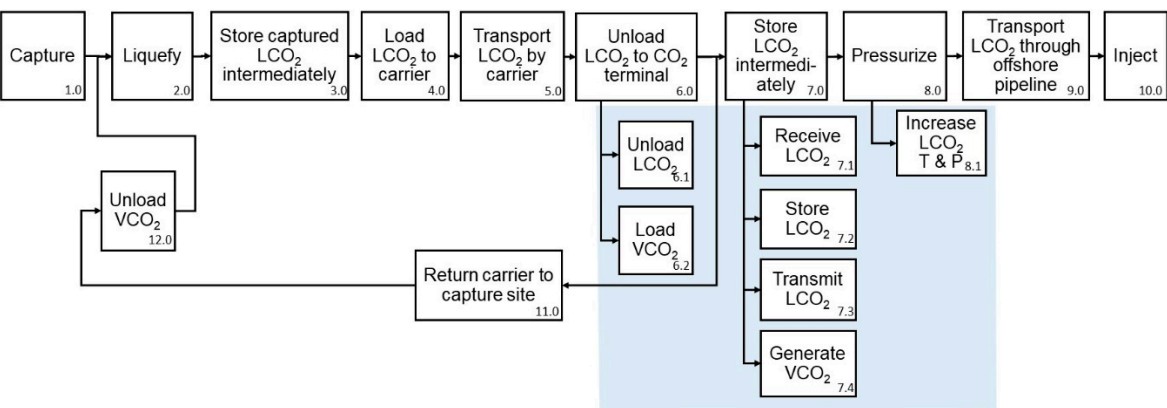

**Figure 5.** A functional flow block diagram of CCS project and scope of $CO_2$ terminal functions (blue region).

### 4.2.2. Integration Definition for Function Modeling (IDEF0) of $CO_2$ Terminal

Through IDEF0, input and output between functions can be explicitly diagrammed. IDEF0 clearly demonstrates the correlations between the functions to be performed. These IDEF0 relationships are represented in Figure 6.

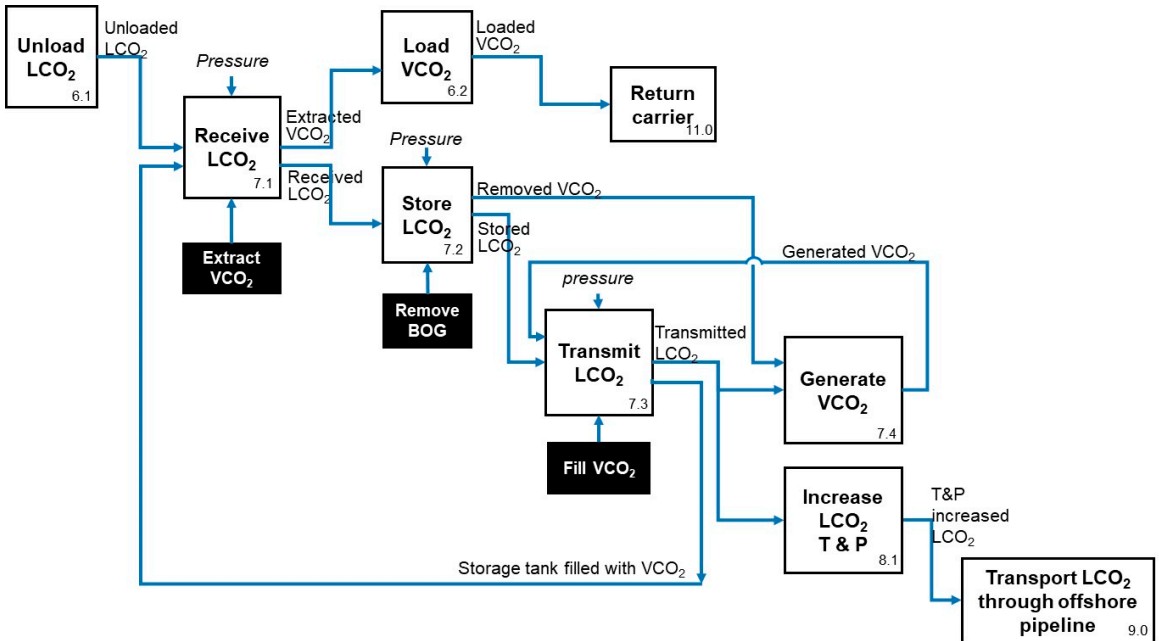

**Figure 6.** IDEF0 of $CO_2$ terminal.

As the input of the system is $LCO_2$ moved by a carrier, the function '6.1 Unload $LCO_2$' is the first function among the seven functions. When the carrier and the $CO_2$ terminal are connected, $LCO_2$ is unloaded from the carrier. The function '7.1 Receive $LCO_2$' is receiving the unloaded $LCO_2$ at the $CO_2$ terminal. Therefore, the output of function 6.1 is 'Unloaded $LCO_2$', which becomes the input of function 7.1. Once $LCO_2$ is received, $LCO_2$ should be retained until transmitted to the offshore pipeline, and this function corresponds to function '7.2 Store $LCO_2$'. To store the received $LCO_2$ in storage tanks, the temperature and pressure of $LCO_2$ should be maintained constantly until $LCO_2$ is transmitted. In this study, we assumed that the temperature and pressure of $LCO_2$ are controlled by extraction of BOG. Therefore, the outputs of the function '7.2 Store $LCO_2$' are 'Stored $LCO_2$' and 'Removed $VCO_2$'.

After, the function '7.3 Transmit $LCO_2$' is completed, which is the transmission of the $LCO_2$ from the storage tanks to offshore pipeline transportation and then the input of function 7.3 is 'Stored $LCO_2$'. When $LCO_2$ is transmitted from the $CO_2$ storage tanks, the pressure in the $CO_2$ storage tanks decreases. To prevent this decrease in pressure, $VCO_2$ should be supplied to the storage tanks and this function corresponds to '7.4 Generate $VCO_2$'. To minimize the wasted $CO_2$, we assumed that input of function 7.4 is the above-mentioned BOG. However, this might not provide a sufficient amount of $VCO_2$. Therefore, we also assumed that the small amount of $LCO_2$ transmitted from the $CO_2$ storage tanks is the second input of function 7.4 and this $LCO_2$ should be vaporized to meet the required $VCO_2$ supply. Once $LCO_2$ transmission is finished, the $CO_2$ storage tank is again ready to receive the $LCO_2$ that is transported by the next carrier.

When conducting '7.1 Receive $LCO_2$', the $VCO_2$ filled in the $CO_2$ storage tanks should be removed to receive $LCO_2$. We assumed that the $VCO_2$ in the $CO_2$ storage tank is loaded into the carrier cargo for the safe return of the carrier to the capture site for the next $LCO_2$ transport. This function corresponds to '6.2 Load $VCO_2$'.

Most of the $LCO_2$ transmitted from the $CO_2$ storage tank is transported via offshore pipelines for injection. Before offshore pipeline transport, the temperature and pressure of $LCO_2$ should be increased appropriately and this function corresponds to '8.1 Increase $LCO_2$ T & P'. Then, the output of function 8.1, 'The temperature and pressure increased $LCO_2$', becomes the input of function '9.0 Transport $LCO_2$ through offshore pipeline'.

### 4.2.3. Timeline Analysis (TLA) of the $CO_2$ Storage Tank Operation

For this study's $CO_2$ terminal, the $CO_2$ is supplied by a $CO_2$ carrier and then transmitted to an offshore pipeline. The $CO_2$ terminal should have enough buffer $CO_2$ to manage $CO_2$ supply interruption. The transmission of $CO_2$ from the $CO_2$ terminal to the offshore pipeline is continuous, whereas the supply of $CO_2$ from a carrier is intermittent. The amount of buffer $CO_2$ should be balanced by both the continuous transmission and the intermittent supply of $CO_2$; therefore, the simple calculation cannot assure the smooth operation of the $CO_2$ terminal without any logistics problems. In this study, TLA, which was explained in Section 3, was used to verify the $CO_2$ logistics supply concept, transmission and buffer amounts, and to determine the storage time of $LCO_2$ and $VCO_2$ per storage tank. In TLA, it is assumed that concurrent filling and transmitting of $CO_2$ in the same tank is forbidden for operational safety reasons.

Figure 7 shows the TLA results. In Figure 7, each cell on the *x*-axis represents four hours, and the *y*-axis represents the seven functions derived in Section 4.1. The activating time of each function is specified by filling the correspondent time square in Figure 7. On the *y*-axis below the functions, rows are additionally inserted to show each $LCO_2$ storage tank state and the arrows represent the transmission of $CO_2$ to the pipeline. Each carrier shipping $CO_2$ is distinguished by filling the time squares with green and blue colors. The yellow represents the initially filled $CO_2$.

In our TLA, we assumed that the $CO_2$ unloading work is conducted for four hours starting from 12:00 every three days. The normal operating period is days 1 to 12. As shown in Figure 7, the seven functions are performed successfully. In a normal operating period, immediately before and after unloading the $LCO_2$, the lowest and highest amounts of buffer $CO_2$ are 7130 and 15,212 tons, respectively. During a three-day cycle, two storage tanks retain $VCO_2$ for 12 and 48 h, respectively, while the other two storage tanks store $LCO_2$. This means that in case of emergency, it is possible to inspect any dysfunctional $CO_2$ storage tank after evacuation of $VCO_2$.

The period from days 13 to 15, marked in red in Figure 7, corresponds to an emergency period where the $CO_2$ carrier arrives at the $CO_2$ terminal 60 h late, but was originally scheduled to arrive on day 13 at noon. As shown in Figure 7, if the unloading of $LCO_2$ from the carrier starts at 20:00 on day 15, it is possible to continuously transmit the $LCO_2$ to the offshore pipeline without any gap. This means four storage tanks enable a buffer of 60 h. For longer buffer hours, additional storage tanks would be required.

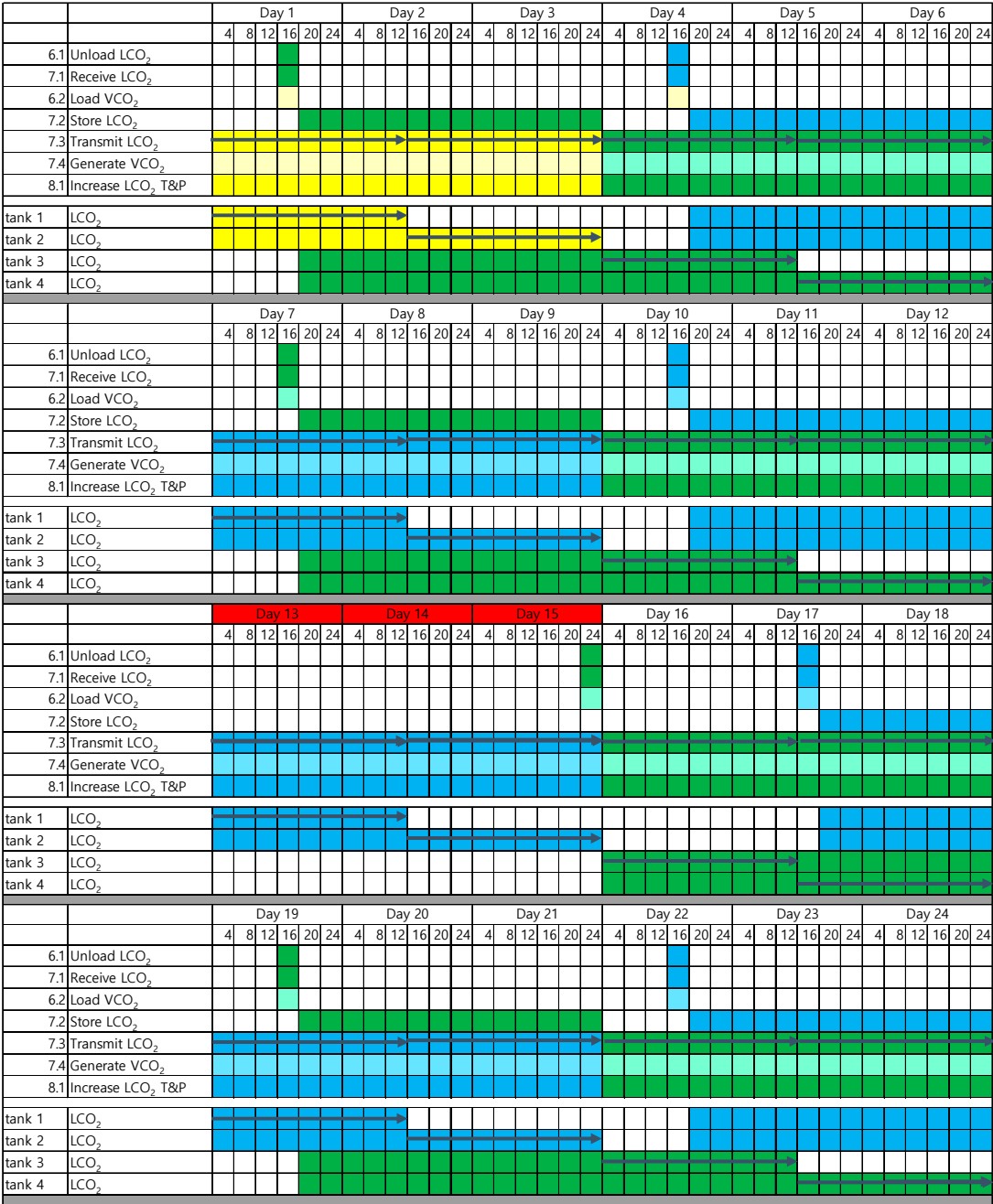

**Figure 7.** Timelines for the activated functions and CO$_2$ storage tanks status. The green and blue colors show each carrier shipping CO$_2$. The yellow represents the initially filled CO$_2$ and the red corresponds to an emergency period.

## 4.3. Identification of Physical Equipment of CO$_2$ Terminal

Identifying the equipment consisting of the system and the performance requirements is important in the system design. In this study, a RAS was used to determine the requirements related to the seven lower-level functions. Firstly, the functions and the related requirements are specified in the left columns in Table 1, which were obtained from Section 2 and the results of Section 4.1. Then, the derived requirements were written in the next column, which were derived from Sections 4.2.1 and 4.2.2. Finally, the necessary equipment was determined and is listed in the right column in Table 1.

**Table 1.** The derived requirements and equipment corresponding to functions of the $CO_2$ terminal.

| No. | Functions | Requirements | Equipment |
|---|---|---|---|
| 5.0 and 11.0 | Transport $LCO_2$ by carrier and Return carrier to capture site | (1) Number of carriers = 2 (2) One-way transport time ≤ 1 day (3) Availability of carriers < 80% (4) Amount of $CO_2$ per one carrier = 8549 tons | Two 9000-ton $CO_2$ carriers |
| 6.1 | Unload $LCO_2$ | (1) Amount of unloaded $LCO_2$ = 8549 tons (2) Time required for unloading ≤ 4 h | $LCO_2$ unloading system |
| 6.2 | Load $VCO_2$ | (1) Amount of loaded $VCO_2$ = 329 tons (2) Time required for loading ≤ 4 h | $VCO_2$ extracting system $VCO_2$ loading system |
| 7.1 | Receive $LCO_2$ | (1) The size of 1 storage tank = 4500 tons | 4500-ton storage tanks |
| 7.2 | Store $LCO_2$ | (1) Temperature and pressure $CO_2$ in storage tank are maintained at −27 °C and 16 bar, respectively, by removing BOG (2) Buffer capacity = 60 h' injection amount (3) The required number of $CO_2$ tanks = 4 | BOG removal system Four 4500-ton storage tanks |
| 7.3 | Transmit $LCO_2$ | (1) Continuous extraction rate of $LCO_2$ from storage tank ≥ 2740 tons/day | LP Pump |
| 7.4 | Generate $VCO_2$ | (1) Daily required amount of $VCO_2$ = 109.7 tons (2) Prevent decrease in pressure of storage tanks by charging $VCO_2$ | $VCO_2$ generating system $VCO_2$ charging system |
| 8.1 | Increase $LCO_2$ T & P | (1) Target temperature = 3–5 °C (2) Target pressure ≥ 120 bar | Heat exchanger Booster pump |

## 5. Discussion

### 5.1. Process Flow Block Diagram of $CO_2$ Terminal

Based on the results obtained from the five analyses, a process flow block diagram of the $CO_2$ terminal was derived (Figure 8). The solid lines and the dotted lines in Figure 8 represent flows of $LCO_2$ and $VCO_2$, respectively. Figure 8 shows that the $CO_2$ terminal consists of three major parts. The first part is the $CO_2$ loading/unloading part (Figure 8, yellow), including an $LCO_2$ unloading system, a $VCO_2$ loading system, and a $VCO_2$ extracting system. According to the TLA result, the $CO_2$ loading/unloading part operates for four hours during a three-day operating cycle. The $VCO_2$ displaced from the storage tanks in the terminal is loaded into the cargo tanks in the carrier. This means the $CO_2$ loading/unloading parts do not require any externally supplied $VCO_2$. The second part is $LCO_2$ transmission (Figure 8, blue), which controls the temperature and pressure of $LCO_2$ for transmission to the offshore pipeline. The corresponding equipment systems are a low pressure (LP) pump, a heat exchanger, and a booster pump. It operates all the time for continuous injection of $LCO_2$. The final part is the vapor-treatment part (Figure 8, red), which is necessary to control the temperature and pressure of the storage tanks for smoothly discharging $LCO_2$ to the offshore pipeline. The most effective method is to recharge the storage tank with $VCO_2$ in the same volume as the discharged $LCO_2$. The vapor-treatment part aims to produce and charge the required $VCO_2$. The vapor-treatment consists of a BOG removal system, a $VCO_2$ charging system, and a $VCO_2$ generating system. In this study, the required $VCO_2$ can be obtained using two methods. The first method involves obtaining $VCO_2$ by BOG from other tanks that are not under discharge. The system responsible for this function is a BOG removal system. The second method involves vaporizing a small portion of the discharged $LCO_2$. This function is handled by a $VCO_2$ generating system. A $VCO_2$ charging system is responsible for injecting $VCO_2$ determined by the above two methods into the storage tanks that discharge $LCO_2$.

The process flow block diagram depicted in Figure 8 helps with understanding the entire $CO_2$ flow and equipment groups. However, with the four $CO_2$ storage tanks and the roles of each tank changing with time, it is difficult to understand the role of each tank using Figure 8 alone. The storage tanks block in Figure 8 is included in all three parts. Then, the storage tanks block is expanded in Figure 8 into four tanks to clarify the different roles of $CO_2$ storage tanks explicitly, as shown in Figure 9.

Figure 9 shows the extended process block flow diagram when storage tanks 1 and 2 are being filled with $LCO_2$ from the carrier during normal operation. This time corresponds to 12:00 to 16:00 on days 7 and 10 in Figure 7. These two $CO_2$ storage tanks belong to the $CO_2$ loading/unloading part (yellow) in Figure 8. Before the carrier has arrived, the two tanks are filled with $VCO_2$ subsequent

to the transmission of $LCO_2$ to the offshore pipeline. After filling $LCO_2$ into $CO_2$ storage tanks 1 and 2, the function of $CO_2$ storage tanks 1 and 2 are converted to the function 'Store $LCO_2$' and the corresponding equipment in Figure 9 is 'storage tank 3'. 'Storage tank 3' belongs to the vapor-treatment part in Figure 8 (red). At this time, the pressure and temperature in the $CO_2$ storage tank are kept constant by extracting the BOG generated in the tank. The equipment responsible for this role is shown in Figures 8 and 9 as 'BOG removal system'. After a period of time, the $LCO_2$ in the tank is then transmitted by the 'LP pump' and the corresponding tank is 'storage tank 4'. 'Storage tank 4' belongs to the $LCO_2$ transmission part in Figure 8 (blue). Most of $LCO_2$ is sent to the 'offshore pipeline' via the 'heat exchanger' and a 'booster pump', and only a small portion of $LCO_2$ is sent to a 'vapor generating system' to generate $VCO_2$. Here, the 'vapor generating system' should also collect the BOG from the 'BOG removal system'. The generated $VCO_2$ is moved into 'storage tank 4' through the 'vapor charging system' to control the pressure of the tank being discharged. $VCO_2$ is ultimately reloaded to the $CO_2$ carrier through a '$VCO_2$ extracting system' and a '$VCO_2$ loading system'.

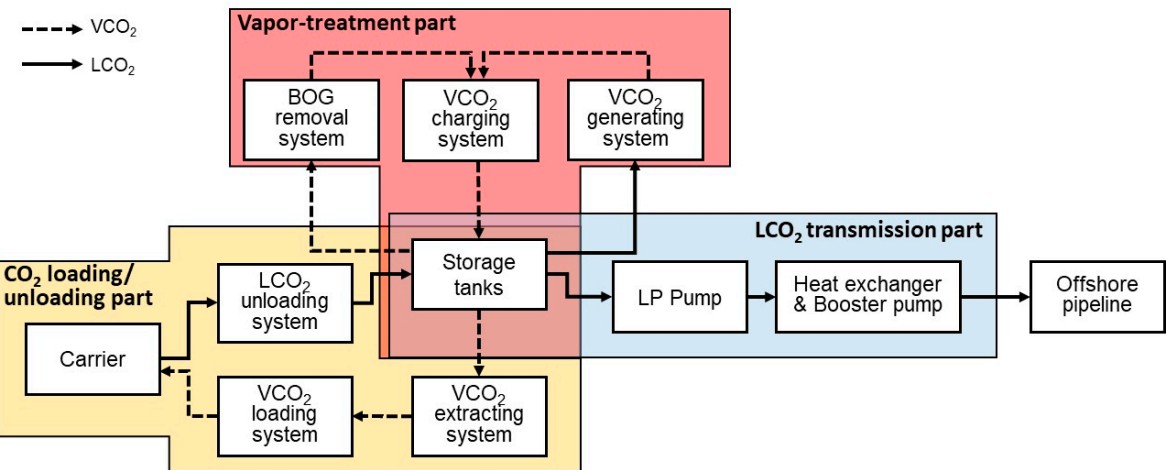

**Figure 8.** The process flow block diagram of the $CO_2$ terminal. The yellow, red, and blue parts represent the $CO_2$ loading/unloading, vapor-treatment, and $LCO_2$ transmission parts, respectively.

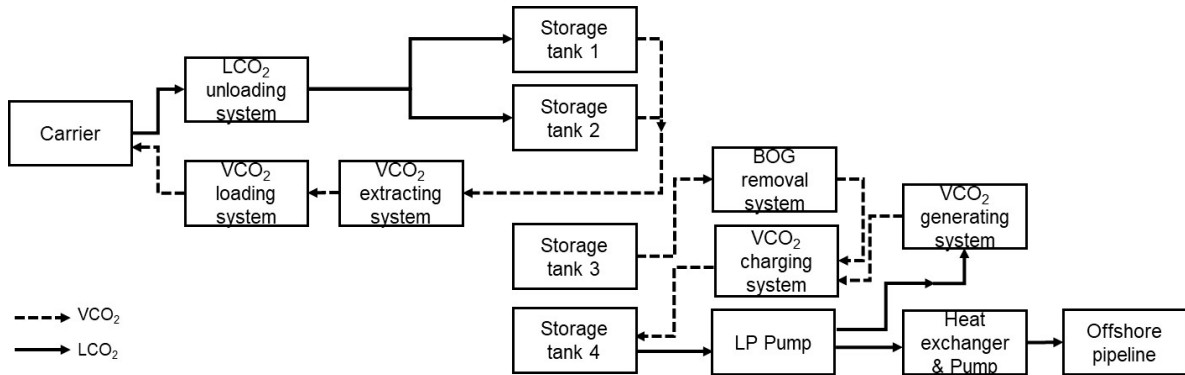

**Figure 9.** The extended process flow block diagram of the $CO_2$ terminal.

### 5.2. Comparison with LNG Terminal

LNG terminals have already been commercialized, with many models available. The concepts $CO_2$ and LNG terminals are similar for temporarily storing carrier-transported liquid in a storage tank and then sending it out through a pipeline. This similarity leads to the mis-prediction that the differences between the LNG terminal and the $CO_2$ terminal are limited to size and operational pressure and temperature conditions. The LNG transport chain and CCS transport chain have considerable differences in transportation distance and transportation pressure and temperature

conditions. The LNG transport chain typically has a long carrier transport distance of 6500 km on average [54] because the LNG production sites are distributed only in a specific area. However, a CCS chain's carrier transport routes are relatively short because offshore $CO_2$ storage sites are widely distributed around the world and the probability of offshore storage sites being near capture sites is high. The temperature and pressure conditions of LNG carriers are usually −162 °C and atmospheric pressure, respectively [55], but the temperature and pressure conditions of $CO_2$ carriers range between the triple point of −56 °C and 5.1 bar and the critical point of 31 °C and 74 bar. Based on the results of this study, the differences in the transportation conditions between an LNG chain and a CCS chain lead to the following important differences in terminal design and operation:

(1) Differences in the number of storage tanks due to the size limit: LNG carriers usually travel long distances and convey a large amount of LNG, so this necessitates a large amount of LNG storage. However, relatively less $CO_2$ storage is required in a $CO_2$ terminal. Since the LNG tanks are operated in cryogenic and atmospheric pressure conditions except for small-scale satellite terminals, one LNG storage tank can be as large as ~160,000 m$^3$ [56]. However, the pressure of the $CO_2$ storage tank should be higher than the triple-point pressure of 5.1 bar, meaning the $CO_2$ storage tanks are pressurized tanks, which are hard to manufacture at capacity bigger than around 5000 tons. This means the $CO_2$ terminal requires multiple tanks, and in many cases, the carrier-transported $CO_2$ has to be unloaded into several storage tanks. In this study, the carrier transports ~9000 tons of $CO_2$, which is unloaded into two tanks that are 4500 tons each. Because overfill is one of the common causes of operational accidents [57], the multiple tanks in the $CO_2$ terminal pose a high risk of overfill. Therefore, ensuring a high level of safety is necessary.

(2) Importance of vaporizer: The storage time in an LNG terminal is usually longer than in a CCS terminal due to the longer carrier transport cycle. In this case, the amount of BOG generated in LNG carriers or storage tanks is huge and needs to be re-liquefied at a high cost and requiring considerable amounts of energy [58]. However, the $CO_2$ carrier transportation cycle and temporary storage period in the terminal are relatively short. In our study, the carrier-transported $CO_2$ is sent to the offshore pipeline within six days according to the TLA results. The amount of BOG in the $CO_2$ terminal is less than the required vapor for preventing the decrease in pressure in the $CO_2$ storage tanks as $LCO_2$ is discharged to the pipeline. Therefore, the vaporizer in a $CO_2$ terminal is essential.

*5.3. Cost Comparison between One and Two Carriers*

This study has determined the number of carriers as two in Section 4.1.2 because operating two carriers is advantageous for continuous injection, although operating one carrier is economically more advantageous. If only one carrier is operated, the design of the terminal is also partially changed, so it is meaningful to check the costs according to the number of carriers operated, including the changes in the cost of terminal induced by the number of carriers.

In the case of operating one carrier, the carrier size and the number/size of storage tanks at the terminal were obtained by following the procedures in Sections 4.1.3 and 4.1.4. According to Equation (1), the amount of $CO_2$ transported during one cycle is 13,700 tons, when operating one carrier, which corresponds to the amount $CO_2$ captured for 5 days. Taking into account the additional $VCO_2$ required at the terminal and assuming that 95% of the cargo tank is filled, the carrier size can be determined as 15,200 tons. As mentioned in Section 4.1.4, the maximum size of one storage tank at the terminal is 5000 tons. Therefore, the number and size of storage tanks required for unloading of $LCO_2$ at the $CO_2$ terminal will be four and 3800 tons, respectively. Considering two buffer tanks and one emitting tank, the total number of required storage tanks is seven.

CAPEX for a $CO_2$ carrier was calculated as $210,000,000 × (carrier capacity/155,000 tons) $^{0.65}$ by calibrating the price of LNG carrier of 155,000-ton size. OPEX for a $CO_2$ carrier was calculated as the sum of 5% of CAPEX and fuel costs. Fuel costs for 9000-ton carrier and 15,000-ton carrier were determined as $5500 and $7100, respectively by applying the equation of the daily fuel cost × 365

days × availability. Assuming that only the number and capacity of storage tanks at the terminals are changed depending on the number of carriers, this study only considered storage tank costs at the $CO_2$ terminal. CAPEX of one storage tank was estimated to be 3.46 times the metal price corresponding to the weight of a thin-wall spherical tank. The weight of the thin-walled spherical tank was obtained from its diameter and thickness, and the metal price was assumed to be $1,666 per ton. OPEX is assumed to be 5% of CAPEX.

To make the comparison between costs easier, CAPEX is converted into the capital recovery cost using the capital recovery factor. The capital recovery cost means the annual equivalent cost of CAPEX. The capital recovery factor is 0.061 with a repayment period of 20 years and an interest rate of 2%.

The calculation results are summarized in Table 2. As shown in Table 2, the annual cost of one carrier is $7.2 million, which is 72% of the cost of two carriers. However, if the storage tank cost is included, the total annual cost when operating one carrier increases to 80% of the total annual cost when operating two carriers. This is due to the increased number of storage tanks at the terminal when operating one carrier. The results show that the number of carriers is very closely related to the terminal design. Therefore, to make an accurate economic evaluation, it is necessary to consider the design parameters of carriers and terminal simultaneously.

**Table 2.** Summary of capacity, capital expenditure (CAPEX), and operating expenditure (OPEX) of carrier and $CO_2$ terminal depending on the numbers of carriers.

|  | No. of Carriers | One Carrier | Two Carriers |
|---|---|---|---|
| Carrier | Size of one carrier (tons) | 15,000 | 9000 |
|  | CAPEX ($) | 51.7 M | 76.2 M |
|  | Capital Recovery Cost ($) | 2.8 M | 4.0 M |
|  | OPEX ($) | 4.4 M | 6.0 M |
|  | Annual cost of carrier ($) | 7.2 M | 10.0 M |
| Terminal | No. of storage tanks | 7 | 4 |
|  | Size of storage tank (tons) | 3800 | 4500 |
|  | CAPEX of Storage tanks ($) | 14.9 M | 9.5 M |
|  | Capital Recovery Cost ($) | 1.0 M | 0.6 M |
|  | OPEX of storage tanks ($) | 0.7 M | 0.5 M |
|  | Annual cost of storage tanks ($) | 1.7 M | 1.1 M |
| Total | Sum of annual costs of carriers and storage tanks at terminal ($) | 8.8 M | 11.0 M |

It is still economical to operate one carrier even if the cost of the terminal increases. If the carrier's failure rate can be reduced to very low values, operating one carrier would be a good option. Adopting novel technologies such as prognostics and health management which have been emerging recently, can minimize unexpected failure of carriers.

## 6. Conclusions

In this study, we aimed to derive the configuration and the operational concept of a $CO_2$ terminal that connects $CO_2$ carriers and an offshore pipeline using the SEP. This paper helps understanding a basic concept of a $CO_2$ terminal. In addition it clearly shows how system engineering process is applied. The following points were derived by conducting the SEP:

(1) FFBD was used to identify the seven basic functions of the $CO_2$ terminal: 'Unload $LCO_2$', 'Load $VCO_2$', 'Receive $LCO_2$', 'Store $LCO_2$', 'Transmit $LCO_2$', 'Generate $VCO_2$', and 'Increase $LCO_2$ T&P'. Then, IDEF0 was used to identify the correlation between these functions.

(2) The short and repetitive $CO_2$ carrier transport affects the configuration of the $CO_2$ terminal. The $CO_2$ terminal here is operated with a three-day cycle and at least four 4500-ton storage tanks are needed. The four tanks allow for continuous $CO_2$ transmission to an offshore pipeline even if a $CO_2$ carrier could not arrive at the $CO_2$ terminal due to the bad weather for 2.5 days. This operational concept of a $CO_2$ terminal was verified by the TLA.

(3) A process flow block diagram was derived from the results of the functional analysis and design synthesis. The configuration of the $CO_2$ terminal consists of a $CO_2$ loading/unloading part, an $LCO_2$ transmission part, and a $VCO_2$ treatment part. These results were used in the subsequent design phase.

(4) In this study, $VCO_2$ is required for two purposes. The first is for filling the carrier cargo tank with $VCO_2$ when unloading $LCO_2$ from the $CO_2$ carrier. In this case, the required $VCO_2$ is covered by displaced $VCO_2$ as $LCO_2$ fills the storage tanks in the terminal. The second purpose is for controlling the pressure in the storage tank that is transmitting $LCO_2$ to offshore pipelines. The required $VCO_2$ is supplied from BOG from the other three storage tanks and vaporized $CO_2$ from the small portion of discharged $LCO_2$.

(5) The comparison of our results with an LNG terminal indicated that a vaporizer is important in the $CO_2$ terminal. As mentioned above, the BOG alone cannot meet the required amount for discharging the $VCO_2$ in a storage tank. Therefore, it is necessary to pay more attention to vaporization rather than reliquefaction in the $CO_2$ terminal. The $CO_2$ terminal requires multiple small-sized storage tanks, unlike the LNG terminal.

(6) Major design factors, such as the number and capacity of storage tanks at terminal, change depending on the number of carriers. Therefore, in order to minimize costs, the design parameters of the carrier and the terminal must be considered simultaneously.

**Author Contributions:** Conceptualization, H.N. and K.K.; methodology, H.N.; validation, C.H. and S.J.H.; formal analysis, K.K.; investigation, H.N.; data curation, C.H.; writing—original draft preparation, H.N.; writing—review and editing, K.K.; visualization, H.N.; supervision, K.K.; project administration, H.K.; funding acquisition, S.-G.K.

**Funding:** This research was supported by a grant from Endowment Project of "Technology development of material handling and risk management for operation and maintenance service of offshore plant" funded by the Korea Research Institute of Ships and Ocean engineering (PES3081).

**Acknowledgments:** The authors are grateful for the full support from the Korea Research Institute of Ships & Ocean Engineering (KRISO).

**Conflicts of Interest:** The authors declare no conflict of interest.

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
