# Peer review of "Conceptualization of CO2 Terminal for Offshore CCS Using System Engineering Process"

_energies, doi:10.3390/en12224350_

Round 1
Reviewer 1 Report
General comments
The quality of the article is not of a high enough standard to accept for publication. The article is interesting and brings up new aspects. However, the language is poor, and I’m unsure if the methods suggested are relevant. A much deeper analysis is required to be able to draw the main conclusion given in line 31 (Abstract): «The biggest difference between CO2 terminal and a LNG terminal is that a vaporizer is necessary in the CO2 terminal rather than a liquefaction package.»
There are numerous grammatical errors through the document e.g. The writing style is, in some places, inappropriate for a scientific journal publication e.g. line 92 «Then, we figure out the functions to meet…». The writing is in some places confusing e.g. from line 175 «Then we estimate the number of carriers and the size of a carrier and a storage tank. Since they highly effect the capital expenditures, it is very important to minimize them.” what is to be minimized? the number or size of the carriers or storage tanks? The intent of this sentence becomes clear when the reader moves to the next sentence, but this does not change the fact that lead sentence for this paragraph is not presented in an understandable way. In total there are far too many errors to correct individually and a comprehensive review should be conducted before the article could be considered for publishing.
I e.g. suggest following corrections to the abstract (line 20-33): «We define equipment that will perform the functions and finally …» Rewrite this sentence (I also recommend not to use the word “We” in the text) «The CO2 terminal in this study is consist of three parts» Remove “is” Line 62 «Unfortunately, actual CCS projects that utilize CO2 terminals are very few, and there has been minimal research regarding the configuration, functions, and roles of CO2 terminals.” This does not accurately reflect the state of knowledge regarding CO2 terminals… Additionally, the ones given, ref 12 & 13, are from 2011, 2014 and 2012. Better referencing/description is required. Section 3 presents only high level concepts relating to the method and gives little detail of the method used in this study. More detail is needed here. Line 72: The abbreviation «BOG» needs to be explained the first time it is used. Line 191 «Therefore, we determined that one carrier transports 8220 tons” is this a realistic tanker size? Add references to support this part of the basis. Line 194 «The same volume of VCO2 must be loaded to the carrier’s cargo tank to prevent negative pressure when unloading LCO2 from the carrier to the CO2 terminal.” I do not understand why this would not be displaced from the vapor space of the storage tanks at the terminal as the fill with liquid CO2 from the carrier. Why does new vapor need to be generated, surely this is just inefficient. Please explain why a vapor generating system at the terminal is better, and explain in more detail how the “BOG removal system”, “VCO2 charging system” and “VCO2 generation system” operates. I also believe a stronger justification is needed of why the methods suggested are assumed to be the best. Please explain exactly what is meant by “negative pressure”, which is used 7 times. Line 391, at the end of discussion: “The amount of BOG in CO2 terminal is smaller than the required vapor for preventing negative pressure of the CO2 storage tanks. Therefore there is a high probability that a CO2 terminal requires a vaporizer, not a reliquifier.” Again, what about all the CO2 vapor in the storage tanks at the terminal which are about to be filled with liquid CO2, why can't that be used…6. Line 232 “The temperature and pressure of LCO2 extracted from the storage tanks needs to be increased to meet an appropriate condition for injection before transmitted through offshore pipeline.” The conventional approach to reaching pipeline pressure would be to use a pump; the authors do not justify the approach that they propose here or mention the simple alternative of using a pump. After pumping, the CO2 is still, of course, at low temperature which can be exploited in the re-liquefaction of BOG at the terminal. No vapor generation is needed as the CO2 is above critical pressure when it is at the pipeline pressure. I do not understand why none of this is discussed. (Line 324 in the discussion chapter is the first place where pumping to pipeline pressure is mentioned in the design of the terminal.)
Reviewer 2 Report
The manuscript is focused on a conceptual design of a CO2 terminal structurally and systematically developed by using the Systems Engineering process (SEP). The article is well written and in the introduction is a complete presentation of the problem of the Carbon Capture and Storage. In this work the validation of the model is based on a specific implant of 1 million of ton and a distance from capture source to CO2 terminal of 580 km. would be interesting that the data reported will be referred also at other type of implant, different for dimension and number of carriers.
Reviewer 3 Report
figures should be improved. you have to underline the novelty of your work in the abstract, introduction and conclusions more references cna be added. You can arrive to about 50 references. check the english languageAuthor Response
Point 1: Figures should be improved.
Response 1: Figure 2(b), Figure 4 and Figure 5 have been improved with clear font size and thick arrows. In addition, in order to help understanding Figure 7, the name of three colored parts have been specified in color boxes in Figure 7.
Point 2: You have to underline the novelty of your work
Response 2: In response to reviewer’s concern, this study’s novelty is underlined in the revised abstract and conclusion, respectively. Please see the revised abstract and conclusion.
Point 3: In the abstract, introduction and conclusions more references can be added. You can arrive to about 50 references.
Response 3: A number of references are added, so the number of references are 53.
Point 4: Check the English language
Response 4: In response to reviewer’s concern, English language has been revised by authors in many places. After that, the manuscript has been checked by English editing services through MDPI (https://www.mdpi.com/authors/english).
Round 2
Reviewer 1 Report
General comments to new article draft:
The quality of the article is still not of a high enough standard to accept for publication. In total there are still too many errors to correct individually, and a much deeper analysis must be required. The article is not organized properly (it could have been half the size without cutting useful information) and the “Functional Analysis” introduced before discussing the “obvious equipment needed” seems to be unnecessary complicated for such a simple problem and make the article much longer and harder to read. Modeling assumptions are weak, and the modeling depth doesn’t get much more complicated than the calculations shown in Eq. 1: “1 × 106 tons / (365 days × 80%/4 days)/2 carriers ≈ 6849 tons/carrier”.
In the previous review round, it was said that “The quality of the article is not of a high enough standard to accept for publication … , the language is poor, … and a much deeper analysis is required to be able to draw the main conclusion …” However, the article and language have been improved significantly (and some serious language/modelling/notation/spelling flaws in the article have been fixed), but there are still plenty of mistakes. For example, just look at the lines 192-198:
“This CCS chain aims to store 1 Mt of CO2 annually, which can be converted into a daily transport rate of 2740 tons/day”… “Typically, large carriers have smaller capital expenditure (CAPEX) and operating expenditure (OPEX) per load than small-sized carriers. Therefore, operating a small number of carriers is cost effective. In other words, one carrier is much cheaper than two half-sized carriers. For example, CAPEX for 10000-ton and 20000-ton K CO2 carriers…”
Three different ways of writing ton have been used in these few sentences: t/tons/ton “K” in “20000-ton K CO2 carriers” should be explained when introduced… (In Table 1 it is written: “The size of 1 storage tank = 4.5 K». It does not look like “K” means exactly the same thing both places…) “Cheaper” sound like a word you should not use in an article… (use e.g. “is less expensive” or “is inexpensive”). It is difficult to see if you mean “large carriers have smaller capital expenditure (CAPEX)”, or “large carriers have smaller capital expenditure (CAPEX) per load”. Do you mean smaller CAPEX/(Volume of the carrier)? “per load” sounds to me more like “per trip”…
Other comments:
Line 199: “Even though it is cheaper to operate one carrier, we determined the number of carriers as two since repair and maintenance of carriers are required. In addition, using two carriers is advantageous to achieve continuous injection of CO2 in the CCS project.”
- This is the fundament for almost all the calculations in this article, and it seems very weak. Can one of these carrier manage to deliver the required LCO2 if the other carrier is repaired? Why is it not possible to achieve continues CO2 injections with one carrier twice as big? (Why not consider a larger total terminal tank volume? This would also reduce the VCO2 that has to be generated, since more BOG will be generated!). Based on this, I’m not sure how interesting the timeline analyses shown in Figure 6 is… (It is also possible that one even larger carrier could be used to transport CO2 from different CO2-sources to the same terminal? Why not build the terminal with a larger total tank volume, so that the unit can still operate if the carrier has to be repaired a few days?)
A deeper economic evaluation is needed, it looks strange to back up some statements from an economic perspective (such as carrier size verses CAPEX and OPEX), but not others (such as terminal tank size versus number of carriers and carrier size). Which is necessary to know in order to find out if using two carriers is optimal in an economic perspective?
A much deeper analysis is required to claim that a vaporizer is essential (for LCO2, but not for LNG terminals): “The biggest difference between the CO2 terminal in this study and the LNG terminal is that a vaporizer is essential in the CO2 terminal due to the smaller storage capacity of the CO2 terminal and, therefore, the lower amount of boil-off gas.”
- This is the main finding that is reported, but there is no modeling of the amount of BOG generated. In order to claim this, the amount of BOG produced should be calculated/estimated, also for different scenarios than the one studied here to see if this is always correct for CO2 terminals. The amount of BOG depends on parameters such as the ambient temperature, insulation of the storage tank, the surface of the multiple small-sized storage tanks suggested for CO2, and other parameters like tank sizes, carrier sizes, transportation distances and shipping schedule e.g. described in Figure 6…
Round 3
Reviewer 1 Report
The paper is revised according to my comments. Thus it is recommended that the paper can be accepted and published.
This manuscript is a resubmission of an earlier submission. The following is a list of the peer review reports and author responses from that submission.